# Synthesis and Characterizations of Na_4_MnCr(PO_4_)_3_/rGO as NASICON-Type Cathode Materials for Sodium-ion Batteries

**DOI:** 10.3390/polym14194046

**Published:** 2022-09-27

**Authors:** Bing-Hsuan Hsu, Wei-Ren Liu

**Affiliations:** Department of Chemical Engineering, R&D Center for Membrane Technology, Center for Circular Economy, Chung Yuan Christian University, 200 Chung Pei Road, Chungli District, Taoyuan City 320, Taiwan

**Keywords:** NASICON, reduced graphene oxide, sodium ion battery, Na_4_MnCr(PO_4_)_3_, cathode

## Abstract

NASICON-type Na_4_MnCr(PO_4_)_3_ (NMCP) wrapped with reduced graphene oxide (rGO) was synthesized via a simple sol-gel method as composite cathode material Na_4_MnCr(PO_4_)_3_/rGO (NMCP/rGO) for Na ion batteries. The surface morphology, crystal structure and pore size distribution of pristine NMCP and as-synthesized NMCP/rGO composite cathode are identified by X-ray diffraction (XRD), field emission-scanning electron microscopy (SEM), transmission electron microscope (TEM), the Brunauer–Emmett–Teller (BET) method and X-ray photoelectron spectroscopy (XPS). The electrochemical performance of composition-optimized NMCP/rGO composite cathode presents stable capacity retention and rate capability. The capacity retention of as-synthesized NMCP/rGO composite is 63.8%, and average coulombic efficiency maintains over 98.7% for 200 cycles. The reversible capacity of as-synthesized NMCP/rGO composite cathode still retained 45 mAh/g and 38 mAh/g under a current density of 0.5 A/g and 1.0 A/g, respectively, which was better than that of pristine NMCP, with only 6 mAh/g and 4 mAh/g. The redox reactions of pristine NMCP and as-synthesized NMCP/rGO composite are studied via cyclic voltammetry. The improved electronic conductivity and structure stability of bare NMCP is attributed to the contribution of the rGO coating.

## 1. Introduction

With the electrification of global technology development, wearable devices and high-tech electronic products are growing demands for energy storage devices. Among these energy storage devices, Li ion batteries are one of the most popular devices because of the merits of long cycle life, higher working voltage, high energy density, high working density and fast charge/discharge [1,2,3,4,5]. The commercialization of lithium-ion batteries has made lithium-ion batteries popular and contributed to a great leap forward in sustainable energy. Therefore, lithium also faces the problems of energy shortage. Scientists are looking for other cheaper and abundant elements to replace Li metal in the future, such as Mg, Ca, K and Na. In addition, their ionic radii are similar to that of lithium ions [6,7,8,9,10,11].

Among the alternative elements, sodium ion batteries have drawn considerable attention due to their merits of high resource content of precursors, wide geographical distribution and low cost [12]. There are many kinds of structure, such as organic salt crystals/mineral crystals [13], polyanionic polymer metal compounds [14] and sodium superionic conductors (NASICONs) [15,16]. The electrochemical properties of polyanionic and NASICONs, such as Na_4_MnTi(PO_4_)_3_ [17], Na_3_V_2_(PO_4_)_3_ [18], Na_4_MnV(PO_4_)_3_ [19] and Na_4_MnCr(PO_4_)_3_ [20,21,22,23] have been studied and reported. NASICON-type materials constitute an important family of polyanionic compounds, which have attracted considerable attention due to their stable 3D framework structure and structural stability. These NASICON-based materials have good thermal stability and a high charge-discharge platform, and they are beneficial to the insertion and de-insertion of sodium ions, which are potential cathode materials for Na-ion batteries. Thus, they have been regarded as the highest energy-density and power-density cathodes for sodium ion batteries [20,21,22,23,24]. Although the ion diffusion rate of NASICON-based materials is fast, its conductivity is not very high; as such, it has poor electrochemical performance. One of the efficient strategies to enhance the electrochemical performance of these cathode materials is to use graphene coating. Graphene is regarded as a potential candidate as an auxiliary material. Due to the advantages of being light weight, with a thin thickness, strong rigidity, and ultra-fast electron transport [5,25,26,27], graphene has been widely used in many studies for energy-related applications, such as fuel cells [28], dye-sensitized solar cells [29], lithium ion batteries [30], sodium ion batteries [31] and so on [32,33]. The structural stability and electrical conductivity of the cathode or anode materials could be improved by the properties of graphene. For example, regarding the application of sodium-ion batteries, Chen et al. [34] reported a spray-drying method to synthesize Na_3_V_2_(PO_4_)_3_/reduced graphene oxide (NVP/rGO) as a cathode material for sodium ion batteries. The disordered band (D band) and graphite band (G band) at 1300~1500 cm^−1^ were observed by Raman analysis, which found that the lower I_D_/I_G_ ratio, the higher the degree of graphitization, which caused the better conductivity. Nyquist plots showed that the NVP/rGO composite cathode exhibited a lower resistance of charge transfer of 399 Ω and a sodium ionic diffusion coefficient of 1.24 × 10^−13^ cm^2^/s, which was higher than the resistance of charge transfer of 716 Ω and the sodium ionic diffusion coefficient of 8.72 × 10^−14^ cm^2^/s of NVP/C. This proved that the ionic conductivity of NVP/rGO becomes better than that of NVP/C. Guo et al. [35] prepared NaVPO_4_F/C/rGO (NVPF/C/rGO) nanocomposite by a sample freeze-drying method. NVPF was made up of nano-sized particles ranging from 200 nm to 500 nm. It was coated by amorphous carbon to form NVPF/C, and then was covered by an rGO sheet on NVPF/C to obtain the NVPF/C/rGO composite. The initial discharge capacity of NVPF/C/rGO was 106.3 mAh/g at 0.05 C. After cycling rate tests from 0.05 C to 10 C, the specific capacity of NVPF/C/rGO nanocomposites could return to 105.8 mAh/g at 0.05 C. The previous studies indicated that rGO can improve the electronic conductivity of NVPF effectively. Kumar et al. [36] performed a typical sol-gel method to synthesize Na_4_MnV(PO_4_)_3_/rGO (NMVP/rGO) composite. The rGO layers connected Na_4_MnV(PO_4_)_3_ (NMVP) particles into a three-dimensional conductive network. After 100 cycles at 0.2 C, the capacity of the NMVP/rGO composite was still 92 mAh/g, and the capacity retention exceeded 90%. Its specific capacity was about 65 mAh/g at an extremely high rate of 20 C, indicating NMVP/rGO exhibited excellent cycle performance and rate capability.

In this work, we synthesized NMCP/rGO composite using a sol-gel method. Afterwards, a series of morphological analyses were characterized by SEM and TEM. The combination of Mn^2+^/Mn^3+^ and Mn^3+^/Mn^4+^ redox couples in the NMCP/RGO composites and prototype NMCP were investigated by cyclic voltammetry at 1.4~4.3 V. The results indicated that NMCP/rGO composite was a potential candidate as a cathode material for Na ion batteries.

## 2. Materials and Methods

### 2.1. Graphene Oxide Preparation

In this study, the Hummers’ method was used to prepare graphene oxide. First, 4 g of NaNO_3_ (99%, Alfa Aesar, Ward Hill, MA, USA) was mixed with 560 mL of H_2_SO_4_ (98%, SHOWA, Alfa Aesar, Ward Hill, MA, USA) and stirred at 80 °C until dissolved. Then, 8 g of graphite (99%, Timcal^®^, GCC Maxwave, Taoyuan city, Taiwan) was added, and the stirred for 2 h, then the temperature was lowered to 10 °C in an ice bath, and 32 g of KMnO_4_ (≥99.5%, SHOWA, Akasaka, Minato-ku, Tokyo) was slowly added to react for another two hours. Then, 800 mL of DI-water dispersion solution was slowly added, followed by 200 mL of H_2_O_2_ (35%, SHOWA, Akasaka, Minato-ku, Tokyo) until the solution turned to yellow. Then, 400 mL of deionized water was added to dilute the solution, and it was left to stand for 3–5 days for graphene oxide (GO) precipitation. Then, the supernatant was removed, 400 mL of HCl (37%, Aencore, Box Hill, Vic 3128, Australia) was added, and then deionized water was added to make the solution up to 2 L; this step was repeated several times until the SO_4_^2−^ ions of the solution were completely replaced by Cl^−^ ions. Finally, the dispersion was dialyzed to pH 7.0, and the product was dried in a vacuum oven after dialysis to neutrality.

### 2.2. Synthesis of Na_4_MnCr(PO_4_)_3_/rGO

As-synthesized NMCP/rGO was prepared using a sol-gel method and a high temperature sintering method (Figure 1). At the beginning, the sieved 5 wt.% GO powder was put into DI-water, and the GO dispersion was formed by shaking it for 1 h in an input shaker. After the temperature of the GO dispersion reached 80 °C, 1 mmol chromium nitrate nonahydrate (Cr(NO_3_)_3_·9H_2_O, 98.5%, Alfa-Aesar, Ward Hill, MA, USA) completely dissolved in 100 mL DI water was added, and the temperature was maintained at 80 °C with continuous stirring. Then, 1 mmol citric acid (C_6_H_8_O_7_·H_2_O, 95%, Sigma-Aldrich, Burlington, MA, USA), 1 mmol manganese acetic acid tetrahydrate ((CH_3_COO)_2_Mn·4H_2_O, ≥99%, Acros-Organi, Janssen-Pharmaceuticalaan, Geel, Belgium), 4 mmol anhydrous sodium acetate (CH_3_COONa, 98%, Alfa-Aesar, Ward Hill, MA, USA), 1.25 g ascorbic acid (C_6_H_8_O_6_, 99%, Acros-Organi, Janssen-Pharmaceuticalaan, Geel, Belgium), and 1.025 mL phosphoric acid (H_3_PO_4_, 86%, Echo Chemical Co., Ltd., Miaoli, Taiwan.) were added, and stirred at 80 °C for 1 h, dried in a circulating oven at 120 °C, and finally sintered at 750 °C under argon atmosphere for 9 h. To prepare pristine NMCP, the same procedure described in the previous part was used, only without the step of adding GO in the preparation process of NMCP/rGO.

### 2.3. Characterizations

The phase purity and crystal structure of pristine NMCP and as-synthesized NMCP/rGO composite were studied by X-ray diffraction (Bruker, D2-Phaser, Billerica, MA, USA) with CuKα radiation (λ = 1.5406 Å) at room temperature. The surface morphologies of as-prepared NMCP and NMCP/rGO composite were observed by field emission-scanning electron microscopy (FE-SEM, JEOL, JSM-7600F, Tokyo, Japan) and transmission electron microscopy (TEM, JEOL, JEM2000FXII, Tokyo, Japan). The pore size distribution and specific surface area of NMCP and NMCP/rGO composite were carried out by the Brunauer– Emmett–Teller equation (BET, Tristar 3000, Norcross, GA, USA). The chemical valence states of sodium, manganese, and chromium in the NMCP/rGO composite were studied by X-ray photoelectron spectroscopy (XPS, Thermo Fisher, K-Alpha, Waltham, MA, USA).

### 2.4. Electrochemical Measurements

Working electrodes were prepared by blending 70% active material (bare NMCP and NMCP/rGO), 20% conductive carbon (Super P), and 10% polyvinylidene difluoride (PVDF) as a binder in solvent (N-Methyl-2-pyrrolidone, NMP) to form the slurry, and then coated on aluminum foil. The electrodes were dried at 120 °C in vacuum for 8 h. The electrolyte was prepared by using 1.0 M NaClO_4_ in ethylene carbonate (EC), propylene carbonate (PC) (1:1 in volume), and 5% fluoroethylene carbonate (FEC) as an electrolyte additive. Sodium foil and glass fiber membranes were applied as the counter electrode and separator for SIBs. Half-cell fabrication took place in an argon-filled glove box for the assembly into CR2032 coin cells. The half-cells were cycled between a potential window of 1.4–4.3 V vs. Na/Na^+^ under galvanostatic conditions by using a battery testing system (Neware, Kowloon Bay, Hong Kong, A211-BTS-4S-1U-ZWJ). Electrochemical impedance spectra (EIS) over the frequency range from 100 kHz to 100 mHz was recorded on a potentiostat (SP300, Biologic, Novi, MI, USA).

## 3. Results and Discussion

Appendix A presents the XRD patterns of pristine-NMCP and as-synthesized NMCP/rGO (Namely Na_4_MnCr(PO_4_)_3_/reduced graphene oxide). The results indicated that these two samples were pure. The Miller index of NMCP/rGO corresponded to NMCP and had no impurity phase. The grain sizes of pristine NMCP and NMCP/rGO were 43 nm and 33 nm, respectively, calculated using the Scherrer equation. The X-ray diffraction (XRD) pattern of Na_4_MnCr(PO_4_)_3_/reduced graphene oxide (NMCP/rGO) and its Rietveld refinement analysis are given in Figure 2a. The X-ray diffraction (XRD) patterns of pristine Na_4_MnCr(PO_4_)_3_ and its Rietveld refinement analysis are given in Appendix A. The NMCP/rGO can be indexed to a typical rhombohedral NASICON structure in the space group R-3c with lattice parameters of a = 8.912 Å, c = 21.396 Å, and *V* = 1471.612 Å^3^. Appendix A shows the lattice parameter of pristine NCP and as-synthesized NMCP/rGO composite. The crystal structure of NMCP is displayed in Figure 2b. It was constructed as an octahedron of MnO_6_ and CrO_6_ corner-sharing with a tetrahedron of PO_4_, and two different sodium sites with six-fold coordination of Na1 and eight-fold coordination of Na2. The sodium ions positioned at Na1 sites were strongly bound to the coordinated oxygen atoms, and the sodium ions resided at Na2 sites were weakly bound, so that in the process of charging and discharging, the migration of sodium ions was mainly contributed by Na2.

Figure 3a,b shows the surface morphology of pristine NMCP particles and as-synthesized NMCP/rGO composite. Both of them were observed by scanning electron microscopy (SEM). The surface of the NMCP particle was smooth with an irregular shape in the range of 1~2 μm. Figure 3c,d display the surface morphology of NMCP/rGO composite. The surface of NMCP/rGO became rougher than that of pristine NMCP, and the surface pores of NMCP became smaller, indicating that the NMCP was successfully coated by rGO. In addition, we used TG analyses to determine the carbon content of pristine NMCP and NMCP/rGO. As shown in Appendix A, the carbon content (rGO) was about 5 wt.%.

Figure 4a shows the TEM image of NMCP/rGO composite, and the inset in Figure 4a displays the selected area electron diffraction (SAED) pattern. Figure 4b reveals the larger magnification image of NMCP/rGO. The TEM images, shown in Figure 4a, demonstrate that the NMCP particle was covered by reduced graphene oxide. According to the d-spacing of 1.64 nm, 1.51 nm, and 2.57 nm (inset in Figure 4b), which resulted from the (4 −1 8), (5 −1 −6), and (3 −1 4) crystal plane, corresponding to the lattice spacing, the results were consistent with the XRD results shown in Appendix A. As shown in Figure 4b, the carbons coated on the NMCP and the rGO layer wrapped around them can be observed in the larger magnification images. The carbon layer was contributed by ascorbic acid. After that, we conducted TEM-energy dispersive spectroscopy (EDS) mapping of NMCP/rGO, as shown in Figure 4c. The elements sodium, manganese, chromium, phosphorus, carbon, and oxygen were distributed uniformly in the NMCP/rGO composite. In the EDX carbon spectrum of the NMCP/rGO composite material, there were still obvious and dense blocks, indicating that the content of carbon was increased, which proved the existence of rGO in NMCP.

In order to confirm the distributions of elements in valence states in NMCP/rGO, we utilized the XPS analyses. Figure 5a displays the XPS survey of the NMCP/rGO composite. The spectrum revealed the binding energy of C 1s, Na 1s, O 1s, Mn 2p, Cr 2p, and P 2p elements in NMCP/rGO composite. Figure 5b–d shows the XPS narrow spectra of Na, Mn, and Cr elements. The single peak at 1070.88 eV was Na 1s orbital, corresponding to the contribution of sodium monovalent. Due to spin–orbit coupling, the two peaks coupling of Mn 2p were Mn 2p3/2 and Mn 2p1/2, respectively. The binding energy of Mn 2p3/2 and Mn 2p1/2 were 640.88 eV and 652.88 eV, respectively. Through the XPS narrow spectrum of Mn, it was possible to observe the existence of Mn^2+^/Mn^3+^. The core-level spectrum of Cr 2p consisted of Cr 2p3/2 and Cr 2p1/2, where the binding energy was located at 576.98 eV and 586.38, respectively. The spectrum illustrated that Cr^3+^ existed in the NMCP/GO composite.

Here, we use the N_2_ adsorption/desorption isotherms to determine the specific surface area and pore size distribution of pristine NMCP and as-synthesized NMCP/rGO samples. The corresponding results are shown in Figure 6a,b. The curves could be classified to the Type IV hysteresis. The specific surface area was found to be 34.2 m^2^/g for NMCP and 12.6 m^2^/g for NMCP/rGO samples. Obviously, the surface area of as-synthesized NMCP/rGO composite was smaller than that of pristine NMCP. It was speculated that the rGO layer covered on the NMCP surface, which made the surface pores disappear. Figure 6b and the inset in Figure 6b displays the pore size distribution of the pristine NMCP and as-synthesized NMCP/rGO composite. The average pore sizes of the materials were found to be 5.80 and 7.80 nm for pristine NMCP and as-synthesized NMCP/rGO composite cathode samples, respectively. The pore size distribution of NMCP was a micropore structure, and the average pore size was about 6 nm. However, the pore size distribution of the NMCP/rGO composite was unevenly distributed from 2 nm to 100 nm, showing a mesoporous distribution. The beneficial mesopores could facilitate ionic migration by constructing the channels for electrolyte infiltration. The larger pore size increased storage space for sodium ions, resulting in improved reversible capacity. The corresponding BET data are summarized in Appendix A.

Appendix A shows the cyclic voltammetry profiles of NMCP and NMCP/rGO for the first three cycles. For the first cycle, pristine NMCP and as-synthesized NMCP/rGO composite were unstable. Regarding this phenomenon, it was presumed that the electrolyte dissociated and reacted with the electrode, and thus formed the CEI (cathode–electrolyte interphase) layer at the first cycle. Except for the first cycle, the others exhibited extremely high overlap, indicating an excellent reversible performance. There are two pairs of redox peaks located at about 3.46/3.72 V and 4.10/4.30 V, respectively. We conducted the CV measurements under different scan rates to determine the relationship between redox current and the sweep rate. Figure 7a,c displays the CV plots of NMCP and NMCP/rGO at the voltage range from 1.4 V to 4.3 V with a scan rate of 0.1 mV/s to 0.5 mV/s. Both of them had two oxidation peaks and two reduction peaks, which were the redox reactions of Mn^2+^/Mn^3+^ and Mn^3+^/Mn^4+^. Redox reaction of pristine NMCP occurred on 3.403 V/3.786 V(Mn^2+^/^3+^) and 4.086 V/4.3 V(Mn^3+^/^4+^). The peak currents of NMCP were 0.056 mA/−0.071 mA(Mn^2+^/^3+^) and 0.109 mA/−0.498 mA (Mn^3+^/^4+^). For the as-synthesized NMCP/rGO, the redox reaction occurred at 3.403 V/3.786 V(Mn^2+^/^3+^) and 4.086 V/4.3 V(Mn^3+^/^4+^). The peak currents of NMCP were 0.337 mA/−0.4346 mA(Mn^2+^/^3+^) and 0.539 mA/−0.836 mA (Mn^3+^/^4+^). The results pointed out that pristine NMCP and as-synthesized NMCP/rGO reacted at almost the same potential, but the peak currents of as-synthesized NMCP/rGO were almost five times than that of pristine NMCP. Of note, the peak currents of pristine NMCP were similar to previous studies; however, the peak current of our NMCP/rGO composite was higher than that of the previously reported one [20]. We conjectured that this condition was caused by covering reduce graphene oxide with excellent conductivity. Figure 7b,d shows the relationship between the peak currents and the square root of the scan rates of pristine NMCP and as-synthesized NMCP/rGO. In order to determine the ionic diffusion dynamics, we further calculate the diffusion coefficient. The diffusion coefficient could be calculated by the following formula:(1)$$D=\frac{i_{p}{}^{2}}{V{\left(2.69\cdot 10^{5}\cdot n^{\frac{3}{2}}\cdot A\cdot C\right)}^{2}}$$
where $i_{p}$ is the peak current (A), n is the charge concentration, A is the contact area between NMCP and the electrolyte, C is the bulk concentration of sodium in the electrode and V is the scan rate. Average diffusion coefficients of pristine NMCP and as-synthesized NMCP/rGO were 7 × 10^−15^ cm^2^/s and 3 × 10^−13^ cm^2^/s, respectively.

Electrochemical analysis of C-rate, cycling performance, AC impedance of pristine NMCP and as-synthesized NMCP/rGO are shown in Figure 8a–d. The corresponding rata capabilities and life cycles of pristine NCMP and NMCP with different rGO content are displayed in Appendix Aa,b, respectively. The results in Appendix A indicated that 5 wt.% rGO was the best condition. The pristine NMCP and as-synthesized NMCP/rGO composite cathode (5 wt.% rGO) at different current densities demonstrate that electrodes delivered superb rate capabilities at current densities ranging from 0.01 to 1 A/g, as shown in Figure 8a. The pristine NMCP only obtained average capacities of 49.4, 32.6, 21.3, 15.1, 10.2, 5.8 and 3.8 mAh/g at 0.01 A/g, 0.02 A/g, 0.05 A/g, 0.1 A/g, 0.2 A/g, 0.5 A/g and 1 A/g, respectively. When the rate returned to 0.01 A/g, the sustained capacity was 32.0 mAh/g. The rate capability of our pristine NMCP was almost the same as previously reported samples with lower carbon content in the literature [19]. However, the as-synthesized NMCP/rGO exhibited considerably enhanced rate capability with high specific capacities and observed average capacities of 77.8, 69.7, 62.1, 56.4, 51.3, 44.2 and 37.6 mAh/g at 0.01 A/g, 0.02 A/g, 0.05 A/g, 0.1 A/g, 0.2 A/g, 0.5 A/g and 1 A/g, respectively. Even if the current density was returned to 0.01 A/g, the sustained capacity was 71.2 mAh/g. Figure 8b displays the cycling performance and the corresponding columbic efficiency of pristine NMCP and as-synthesized NMCP/rGO; at the higher current density of 0.1 A/g, as-synthesized NMCP/rGO still delivered a specific capacity of 41.0 mAh/g, even after 200 cycles, and the average columbic efficiency was 98.7 %. Pristine NMCP at the high rate 0.1 A/g and after 200 cycles only obtained a specific capacity of 6.6 mAh/g and a columbic efficiency of only 97.3 %. The NMCP/rGO sample shows great enhancement of the capacity retention in comparison with the pristine NMCP sample. The improvement of the NMCP/rGO, in terms of cycling stabilities and high rate performance, may be ascribed to the strong interaction between the NMCP particles and rGO sheets, as well as the good conductivity of rGO.

Figure 8c shows the Nyquist impedance spectra results of pristine NMCP and as-synthesized NMCP/rGO electrode-consisting cells after 2.5 cycles. The electrode kinetic parameters are shown in Appendix A. The corresponding fitted equivalent circuit is shown in the inset of Figure 8c. After rGO coating, the resistance of NMCP from approximately 600 Ω decreased to 100 Ω, indicating that rGO improved the ability of electric/ionic transfer. Figure 8d displays the linear relationships between Z’ and the reciprocal square root of angular rate (ω^−1/2^) in the low frequency region. We further observed the dynamic of ionic diffusion by using the diffusion coefficient which could be calculated from the following formula: (2)$$D=\frac{R^{2}T^{2}}{2\cdot A^{2}n^{4}F^{4}C^{2}\sigma^{2}}$$
where R is the ideal gas constant, T is the Kelvin temperature, F is the Faraday constant, *n* is the charge concentration, A is the contact area between NMCP and electrolyte, C is the bulk concentration of sodium in electrode and σ is the scan rate. Finally, we compared these two results, and found that the diffusion coefficient of as-synthesized NMCP/rGO was ten times over than that of pristine NMCP.

Appendix A show the charge/discharge profiles of pristine NMCP and the as-synthesized NMCP/rGO composite at a current density 0.01 A/g for the first three cycles. At the first cycle, the columbic efficiency of NMCP was only 55.3%, while the NMCP/rGO was 83.2%. The overlap of charge/discharge curves of NMCP/rGO at the 2nd and 3rd cycles demonstrated that the structural stability of NMCP could be dramatically improved by rGO coating.

The charge/discharge profiles of pristine NMCP and as-synthesized NMCP/rGO at various current densities from 0.01 A/g to 1 A/g are displayed in Figure 9a,c. When the current density of NMCP was 0.01 A/g, an insignificant voltage plateau could be observed, and it was difficult to observe the voltage plateau after the current density was greater than 0.02 A/g. It was inferred that the poor structural stability and conductivity of NMCP lead to the collapse of the internal structure, and the ion migration path was destroyed, resulting in concentration polarization. Therefore, rGO was used to stabilize the structure and improve the conductivity of NMCP. Figure 9c represents the charge/discharge profile of as-synthesized NMCP/rGO. All curves were similar to each other. The variation voltage of the platform was caused by polarization of NMCP/rGO. Platforms were still observed due to the enhanced structural stability and electrical conductivity of rGO. The charge/discharge curves of pristine NMCP and as-synthesized NMCP/rGO at a. current density 0.1 A/g for two hundred cycles are shown in Figure 9b,d. The platform of pristine NMCP was hardly observable at a high current density of 0.1 A/g; this phenomenon could be attributed to the occurrence of severe polarization in the NMCP. However, plateaus of as-synthesized NMCP/rGO were obvious; this means the polarization of as-synthesized NMCP/rGO composite cathode was slightly lower than that of pristine NMCP, which was presumably due to the rGO coating. 

The SEM images of pristine NMCP and as-synthesized NMCP/rGO electrodes before and after 200 cycles are shown in Figure 10a,b and Figure 10c,d, respectively. After 200 cycles of charge and discharge at a current density of 0.1 A/g, the complete structure of NMCP electrode had many cracks which greatly reduced the storage space of sodium ions, resulting in a continuous decline in capacity and low capacity retention rate. Under the same test conditions, NMCP/rGO showed almost no traces of damage on the surface of the electrode. Because rGO stabilized the structure of NMCP, it was not easy to collapse due to the insertion and deintercalation of sodium ions during long-term charge–discharge cycle tests. 

## 4. Conclusions

In summary, we successfully prepared pristine NMCP and as-synthesized NMCP/rGO composite cathode by using a sol-gel method. The particles of pristine NMCP were microparticles, in the range of 1~2 μm. The NMCP-electrode retained 26.4% of the initial discharge capacity of 24.6 mAh/g after 200 cycles at 0.1 A/g. After the rate capability tests, the discharge capacity of pristine NMCP reached 3.78 mAh/g at 1 A/g; at 0.01 A/g, it showed 54.0% retention compared to the initial discharge capacity of 57.9 mAh/g. The poor retention and rate performance were improved by rGO coatings. The coated rGO can provide a path for electronic transportation, preventing the structure from being destroyed and thereby improving its electronic conductivity. Based on the above advantages of the rGO coating layer, the composition-optimized NMCP/rGO sample expressed a rate capacity of 37.5 mAh/g at a high current density of 1 A/g. At 0.01 A/g, the sample exhibited 88.1% retention compared to the initial discharge capacity of 80.6 mAh/g. From the long cycling tests, as-synthesized NMCP/rGO exhibited 63.9% retention compared to the initial discharge capacity of 64.2 mAh/g after 200 cycles at 0.1 A/g. The NMCP/rGO electrode showed outstanding cycling performance and superior rate capability. As a result, the sodium storage performance of NMCP was largely improved by rGO coatings. As such, it is regarded as a promising material for improving the electrodes of SIBs.

## Figures and Tables

**Figure 1 polymers-14-04046-f001:**
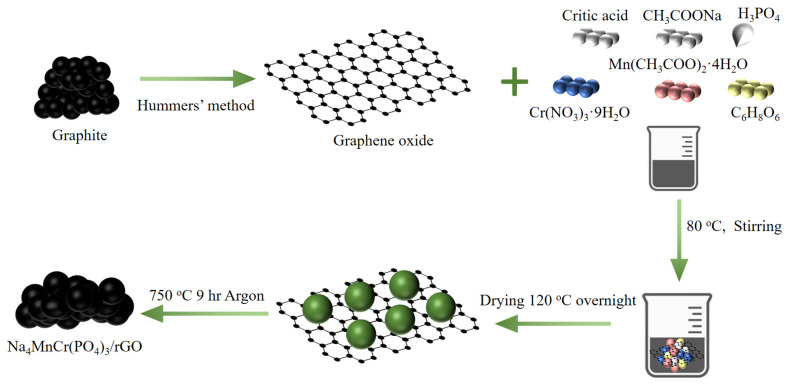
Illustrations of the Hummers’ method and the sol-gel process for preparing Na_4_MnCr(PO_4_)_3_.

**Figure 2 polymers-14-04046-f002:**
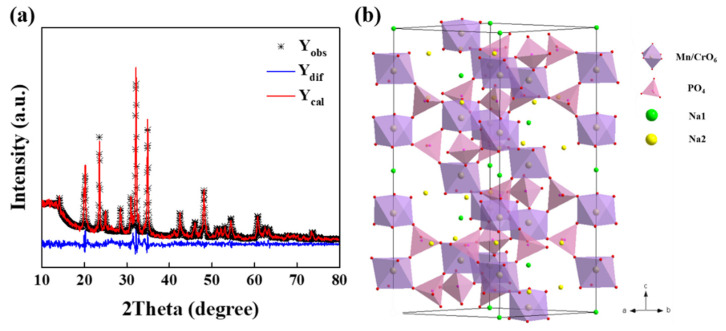
(**a**) X-ray diffraction (XRD) and corresponding Rietveld refinement results of NMCP/rGO; (**b**) crystal structural of NMCP and the coordination of cations.

**Figure 3 polymers-14-04046-f003:**
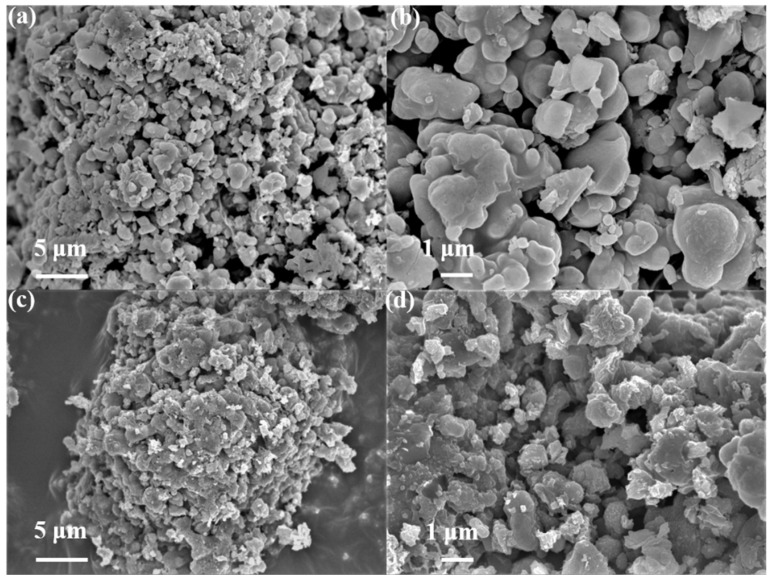
SEM images of (**a**,**b**) pristine NMCP and (**c**,**d**) as-synthesized NMCP/rGO under different magnifications.

**Figure 4 polymers-14-04046-f004:**
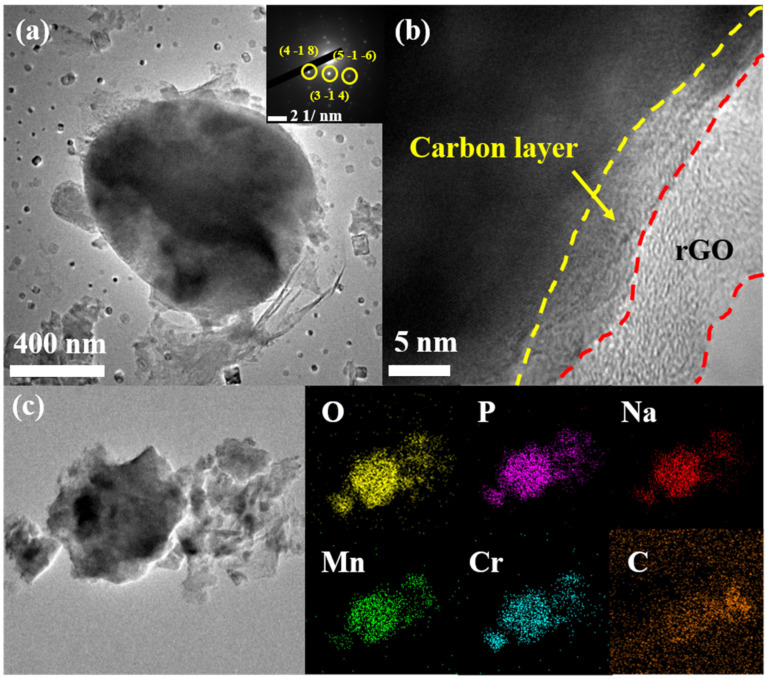
(**a**,**b**) TEM images of NMCP/rGO and SAED; (**c**) NMCP/rGO elemental mappings of oxygen (yellow), phosphorus (purple), sodium (red), manganese (green), chromium (blue) and carbon (orange).

**Figure 5 polymers-14-04046-f005:**
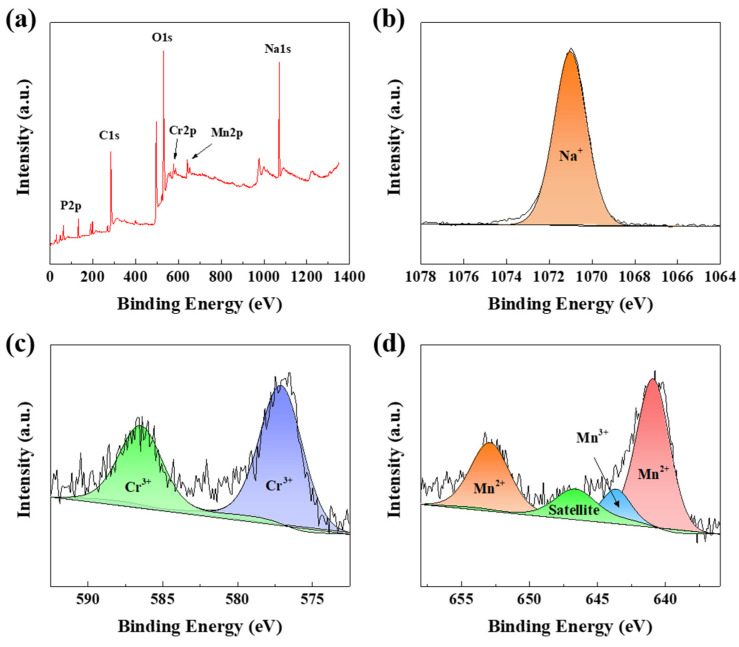
XPS spectra of (**a**) survey, (**b**) Na 1s, (**c**) Cr 2p, and (**d**) Mn 2p of NMCP/rGO.

**Figure 6 polymers-14-04046-f006:**
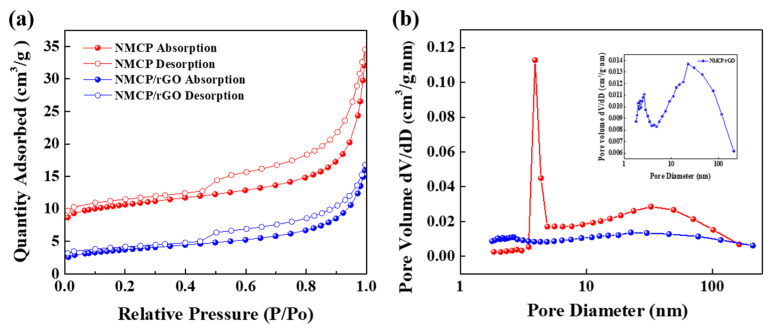
Bare NMCP and NMCP/rGO: (**a**) absorption and desorption isotherms; (**b**) pore size distributions. Inset in (**b**) shows the pore size distribution of NMCP/rGO in large magnitude.

**Figure 7 polymers-14-04046-f007:**
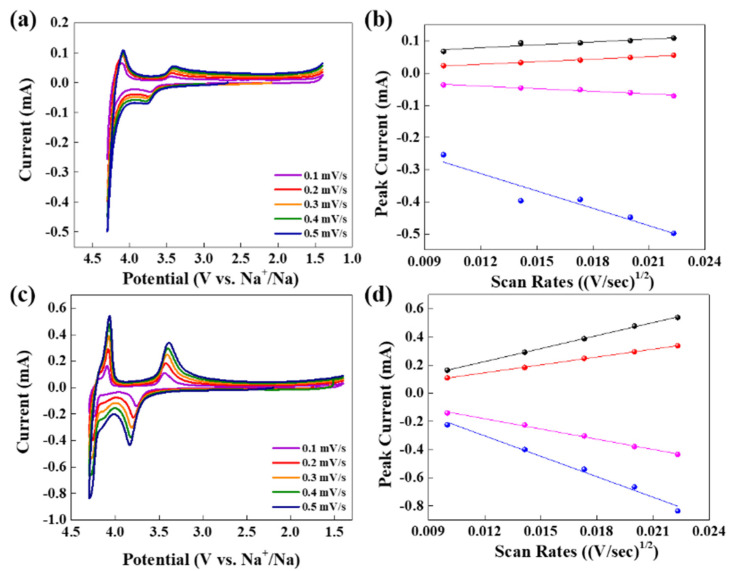
Cyclic voltammetry profiles of (**a**) NMCP and (**c**) NMCP/rGO at 0.1 mV/s to 0.5 mV/s, and the relationship between the peak currents and the square root of the scan rates of (**b**) NMCP and (**d**) NMCP/rGO.

**Figure 8 polymers-14-04046-f008:**
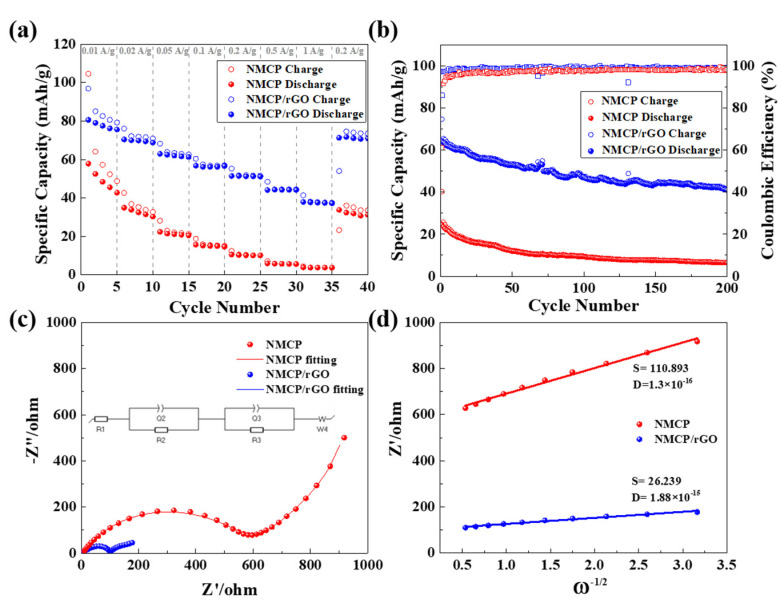
NMCP and NMCP/rGO (**a**) C-rate tests at 0.01 A/g to 1 A/g, (**b**) cycling performances at 0.1 A/g, (**c**) Nyquist plots along with the fitting curves and equivalent circuit, and (**d**) linear relationships between Z’ and the reciprocal square root of angular rate (ω^−1/2^) in the low frequency region.

**Figure 9 polymers-14-04046-f009:**
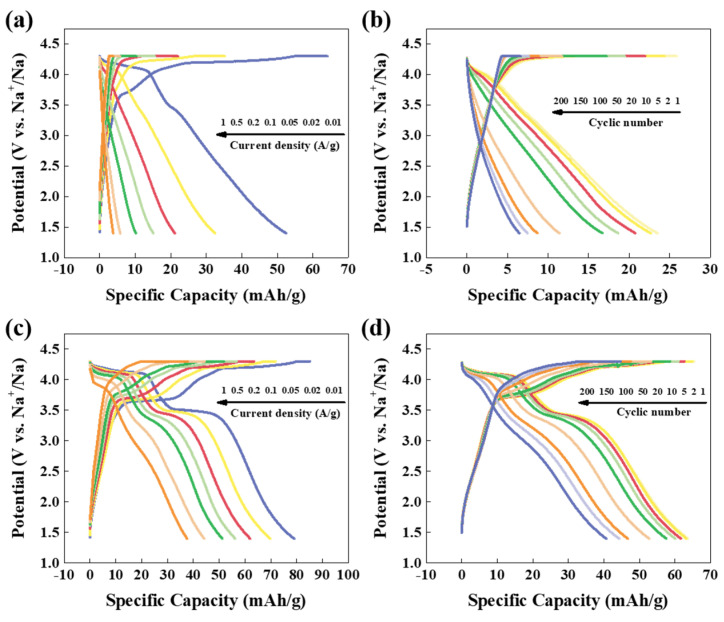
(**a**) Galvanostatic charge/discharge curves of the NMCP electrode at different current rates from 0.01 to 1 A/g. (**b**) Representative charge/discharge curves of NMCP at 0.1 A/g. (**c**) Galvanostatic charge/discharge curves of the NMCP/rGO electrode at different current rates from 0.01 to 1 A/g. (**d**) Representative charge/discharge curves of NMCP/rGO at 0.1 A/g.

**Figure 10 polymers-14-04046-f010:**
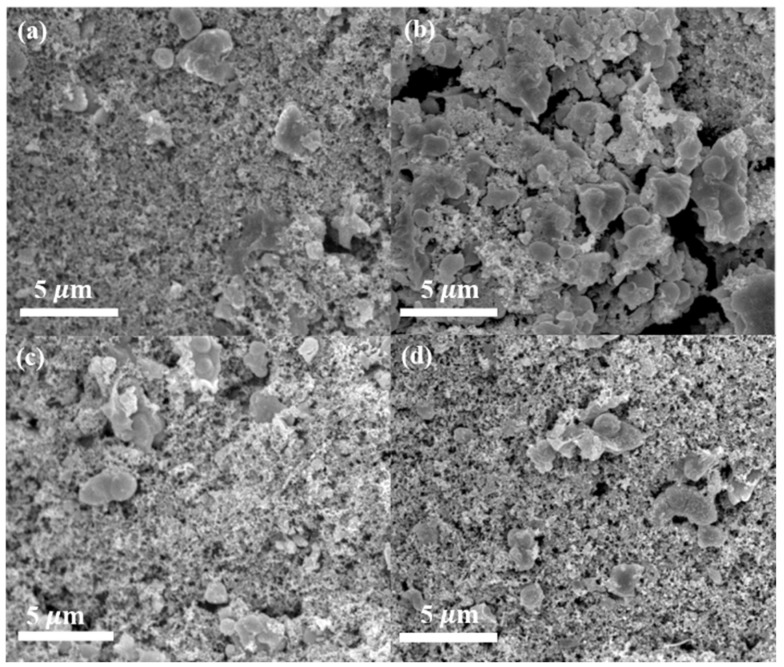
SEM images of the cathode before and after cycling. (**a**) NMCP cathode before cycling; (**b**) NMCP cathode after cycling; (**c**) NMCP/rGO cathode before cycling; (**d**) NMCP/rGO cathode after cycling.

## Data Availability

Not applicable.

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
