# Peer review of "Synthesis and Characterizations of Na4MnCr(PO4)3/rGO as NASICON-Type Cathode Materials for Sodium-ion Batteries"

_polymers, 2022, doi:10.3390/polym14194046_

Round 1

Reviewer 1 Report (Previous Reviewer 2)

Dear Author,

Nice day to you.

The ِAuthor(s) has completed all the necessary corrections required, so I recommend publishing this research in this journal... With Regards

Author Response

The ِAuthor(s) has completed all the necessary corrections required, so I recommend publishing this research in this journal.

Response:

Thank you for reviewer's  positive response.

Reviewer 2 Report (Previous Reviewer 1)

The manuscript has been improved after revision. It can be considered for publication after further revision. 

1. Please carefully check the text and correct the grammatical errors.

2. Figures that provided in the Supporting Information should appear in sequence in the main text.

3. The Figures of electrochemical performance of NMCP/rGO with different rGO content are suggested to be provided in the Supporting Information.

Author Response

1. Please carefully check the text and correct the grammatical errors.

Response:

Thank you for reviewer's suggestions. We have checked the text and corrected the grammatical errors in the revised manuscript.

2. Figures that provided in the Supporting Information should appear in sequence in the main text.

Response:

We have provided this information in the revised manuscript.

3. The Figures of electrochemical performance of NMCP/rGO with different rGO content are suggested to be provided in the Supporting Information.

Response:

Thank you for reviewer's good suggestions. We have provided the electrochemical performance of NMCP/rGO with different rGO content in the revised supporting information.

Reviewer 3 Report (New Reviewer)

I have read the article titled “Synthesis and Characterizations of Na4MnCr(PO4)3/rGO as NASICON-type Cathode Materials for Sodium-ion Batteries” by Liu et al submitted to Polymers MDPI. Energy storage technology is one of the most critical technologies to the development of new energy electric vehicles and smart grids. Among the energy storage technologies, lithium/sodium-ion technologies are an attractive energy storage device exhibiting excellent efficiency of charge-discharge processes along with high energy density among the available other rechargeable systems. Materials with a polyanionic framework, such as NASICON structured cathodes with formula NaxM2(PO4)3, have attracted considerable attention because of their stable three-dimensional crystal structure, and high operating potential. In the submitted work, the authors have reported a composite Na4MnCr(PO4)3/rGO as a promising cathode material for Na-ion batteries.

NMCP itself is not a novel material, it has been reported earlier by Ceder et al, Jun Chen et al . However, NMCP has been synthesized as a composite with rGO using sol-gel. Despite these, why the electrochemical performance is so marginal (about 90 mAh.g)? It requires a prompt justification.

Therefore, the demonstrated work is a significant area of research. The paper itself is written OK but the main concern of this paper and the areas that require clarification are given below.

·         Usually, the Na3V2(PO4)3 can deliver a reversible capacity of about 110 mA h g−1 with a relatively high voltage plateau located at 3.4 V, having a higher energy density of about 370 Wh kg−1.  Why the reported NMCP is very low in this work?

·         What is the number of electron processes involved in the Mn, and Cr couple?

·         What is the synergistic effect of Mn and Cr in the crystal structure of Nasicon? However, only 35 cycles have been shown. During charge and discharge does NMCP endures a lattice change?

·         Considering the different coordination environments of phosphate and oxides, the standard oxides of MnO, Mn2O3, and MnO2 be observed during the redox reactions that hinder the realizing higher discharge capacity?

·         First paragraph of the introduction: Li-ion battery chemistry and its relevant cathode precursor proportion need to be mentioned and referred to in the material reported in the literature (such as Progress in Solid State Chemistry 62 (2021) 100298; doi.org/10.3390/en13061477).

·         What do the EIS fitting (In Fig. 8) results reveal? Does it beneficial for rate capability, due to lower Rct and faster Na-ion diffusivity? Otherwise, what is the purpose of rGO?

·         Please benchmark the electrochemical mechanism and discharge capacity with the materials reported for Na-ion battery in the literature (10.1039/C8NR03824D; doi.org/10.1016/j.mtener.2018.08.004)

·         What is the role of carbon coating from rGO in improving the electrochemical activity of the present NMCP?

·         Page 10; how to improvise the electrode structure without the formation of cracks?

·         The last line on page 10; is the mechanism involves only intercalation/deintercalation or any other phase transformation?

·         In section 4, please indicate the initial charge/discharge capacity and then its retention after 35 cycles for both the pristine and NMCP composites.

Author Response

I have read the article titled “Synthesis and Characterizations of Na4MnCr(PO4)3/rGO as NASICON-type Cathode Materials for Sodium-ion Batteries” by Liu et al submitted to Polymers MDPI. Energy storage technology is one of the most critical technologies to the development of new energy electric vehicles and smart grids. Among the energy storage technologies, lithium/sodium-ion technologies are an attractive energy storage device exhibiting excellent efficiency of charge-discharge processes along with high energy density among the available other rechargeable systems. Materials with a polyanionic framework, such as NASICON structured cathodes with formula NaxM2(PO4)3, have attracted considerable attention because of their stable three-dimensional crystal structure, and high operating potential. In the submitted work, the authors have reported a composite Na4MnCr(PO4)3/rGO as a promising cathode material for Na-ion batteries.

NMCP itself is not a novel material, it has been reported earlier by Ceder et al, Jun Chen et al. However, NMCP has been synthesized as a composite with rGO using sol-gel. Despite these, why the electrochemical performance is so marginal (about 90 mAh.g)? It requires a prompt justification.

Response:

Thank you for reviewer’s comments. Yes. Compared to previous literature, our NCMP cathode is ~90 mAh/g. The reasons might be due to different electrolyte composition, different recipe of electrode, thickness, porosity and so on. Even though “the same” chemical composition of cathode, the reported reversible capacity is different. Our contribution is to provide a modification strategy by using reduced graphene oxide as a conductive matrix to enhance the electrochemical performance of NMCP cathode (first time) for Na ion batteries. The detail materials characterizations and electrochemical of pristine NCMP and NMCP/rGO composite were demonstrated. We believe the manuscript could be a very important reference for developing cathode materials for Na ion batteries.

Therefore, the demonstrated work is a significant area of research. The paper itself is written OK but the main concern of this paper and the areas that require clarification are given below.

  • Usually, the Na3V2(PO4)3can deliver a reversible capacity of about 110 mA h g−1 with a relatively high voltage plateau located at 3.4 V, having a higher energy density of about 370 Wh kg−1.  Why the reported NMCP is very low in this work?

Response:

Thank you for reviewer’s comments. Yes. Compared to previous literature, our NCMP cathode is ~90 mAh/g. The reasons might be due to different electrolyte composition, different recipe of electrode, thickness, porosity and so on. Even though “the same” chemical composition of cathode, the reported reversible capacity is different. Our contribution is to provide a modification strategy by using reduced graphene oxide as a conductive matrix to enhance the electrochemical performance of NMCP cathode for Na ion batteries. The detail materials characterizations and electrochemical of pristine NCMP and NMCP/rGO composite were demonstrated. We believe the manuscript could be a very important reference for developing cathode materials for Na ion batteries.

  • What is the number of electron processes involved in the Mn, and Cr couple?

Response:

Thank you for reviewer’s suggestion. As shown in the reference [R2], for charge process, the voltage above 3.8 V is Mn3+ à Mn4+. The voltage above 4.2V is Cr3+ à Cr4+. In addition, there are two electrons involved in the Mn, and they are occurred on Mn2+/3+ and Mn3+/4+ redox reaction. Because at 4.3 V, the voltage plateau of chromium cannot be generated yet. If we want to get the voltage plateau of chrome, we have to raise the voltage window above 4.3 V.

  • What is the synergistic effect of Mn and Cr in the crystal structure of Nasicon? However, only 35 cycles have been shown. During charge and discharge does NMCP endures a lattice change?

Response:

Thank you for reviewer’s suggestion. The electrons continuously transfer from the Mn-3d state during the first two voltage plateaus before the Cr-3d when the last Na+ is extracted. Charge of Mn mainly increases during the first two plateaus while that of Cr mainly increases during the last plateau. The average Mn–O bond length decreases from 2.203 to 1.983 Å during the first two Na+ extracted, and it changes slightly when the last Na+ is extracted. The average Cr–O bond length is almost unchanged during the first two voltage plateaus, it decreases from 2.001 to 1.968 Å when the last Na+ is extracted.

The lattice changes during charging and discharging, and the volume of the lattice shrinks due to the migration of sodium ions during charging. During discharge, sodium ions re-immigrate to make the lattice volume return to its original size.

[R1]Liu, J.; Wang, S.; Kawazoe, Y.; Sun, Q. Mechanisms of Ionic Diffusion and Stability of the Na4MnCr(PO4)3 Cathode, ACS Materials Letters. 2022, 4, 860-867.

[R2] Wang, J.; Wang, Y.; Seo, D. H.; Shi, T.; Chen, S.; Tian, Y.; Kim, H.; Ceder, G. A High‐Energy NASICON‐Type Cathode Material for Na‐Ion Batteries, Advanced Energy Materials. 2020, 10, 1903968.

  • Considering the different coordination environments of phosphate and oxides, the standard oxides of MnO, Mn2O3, and MnO2 be observed during the redox reactions that hinder the realizing higher discharge capacity?

Response:

Thank you for reviewer’s suggestion. During the charging and discharging process, a phase change occurs to generate compounds. The factor of low discharge capacity comes from conductivity, so coating rGO can solve this problem

[R2]Wang, J.; Wang, Y.; Seo, D. H.; Shi, T.; Chen, S.; Tian, Y.; Kim, H.; Ceder, G. A High‐Energy NASICON‐Type Cathode Material for Na‐Ion Batteries, Advanced Energy Materials. 2020, 10, 1903968.

  • First paragraph of the introduction: Li-ion battery chemistry and its relevant cathode precursor proportion need to be mentioned and referred to in the material reported in the literature (such as Progress in Solid State Chemistry 62 (2021) 100298; doi.org/10.3390/en13061477).

Response:

Thank you for reviewer’s comments. The relevant reference about cathode precursor proportion has been cited and mentioned in the revised manuscript.

[35] Divakaran, A. M.; Minakshi M.; Bahri P. A.; Paul S.; P., Divakaran A. M.; NamaManjunatha K. Rational design on materials for developing next generation lithium-ion secondary battery. Solid State Chemistry 2021, 62, 100298.

  • What do the EIS fitting (In Fig. 8) results reveal? Does it beneficial for rate capability, due to lower Rct and faster Na-ion diffusivity? Otherwise, what is the purpose of rGO?

Response:

The EIS fitting results reveals the detail contribution of impedances are resulted from RCEI or RCT. The rGO mainly improves the electronic conductivity of NMCP and enhances the structural stability of NMCP. Thus, Rct was dramatically reduced after rGO coating. In addition, the rGO coating also facilitate Na ion diffusion.

  • Please benchmark the electrochemical mechanism and discharge capacity with the materials reported for Na-ion battery in the literature (10.1039/C8NR03824D; doi.org/10.1016/j.mtener.2018.08.004)

Response:

Thank you for reviewer’s suggestion. We have cited the important reference in the revised manuscript.

[36] Minakshi M.; Barmi M.; Mitchell D. R.G.; Barlow A. J.; Fichtner M. Effect of oxidizer in the synthesis of NiO anchored nanostructure nickel molybdate for sodium-ion battery. Materials Today Energy 2018, 10, 1-14.

  • What is the role of carbon coating from rGO in improving the electrochemical activity of the present NMCP?

Response:

In this paper, we mainly discuss the effects of rGO. The rGO mainly improves the electronic conductivity of NMCP and enhances the structural stability of NMCP. In our study, both NMCP and NMCP/rGO have carbon layer, the difference is that NMCP/rGO has more rGO content. Thus, the effect of carbon coating cannot be compared with other literature. However, the effects of carbon coatings have been compared in previously work.

  • Page 10; how to improvise the electrode structure without the formation of cracks?

Response:

The SEM images of pristine NMCP and as-synthesized NMCP/rGO electrodes before and after 200 cycles are shown in Fig. 10(a, b) and Fig. 10(c, d), respectively. We know it is not easy to identity the formation of cracks for bare NMCP and NMCP composite because the porous nature of the electrodes. However, please observe the surface morphology change between Fig. 10(a) and 10(b). It is really a large change for bare NCMP electrode after cycling. Without reduced graphene oxide coating and projection, it cannot maintain the structure of electrode during charge and discharge processes. We hope reviewer could satisfy with our response.

  • The last line on page 10; is the mechanism involves only intercalation/deintercalation or any other phase transformation?

Response:

Thank you for reviewer’s suggestion. Following by the previous literature. Na is further extracted by a two-phase mechanism as evidenced by the discrete position change of the (024) and (116) peaks and the disappearance of the (211) and (300) peaks. In addition, a new peak appears at 16.2°. Thus, we can illustrate that charge and discharge will caused the phase change.

[R2] Wang, J.; Wang, Y.; Seo, D. H.; Shi, T.; Chen, S.; Tian, Y.; Kim, H.; Ceder, G. A High‐Energy NASICON‐Type Cathode Material for Na‐Ion Batteries, Advanced Energy Materials. 2020, 10, 1903968.

  • In section 4, please indicate the initial charge/discharge capacity and then its retention after 35 cycles for both the pristine and NMCP composites.

Response:

Thank you for reviewer’s comments. The initial charge/discharge capacity and then its retention after 35 cycles for both the pristine and NMCP/rGO composite have been indicated in the revised manuscript.

“The NMCP-electrode retained 26.4% of the initial discharge capacity of 24.6 mAh/g after 200 cycles at 0.1 A/g. After the rate capability tests, the discharge capacity of pristine NMCP reached to 3.78 mAh/g at 1 A/g, and then back to 0.01 A/g showed 54.0% retention compared to the initial discharge capacity of 57.9 mAh/g.

Round 2

Reviewer 3 Report (New Reviewer)

The revised version is suitable to publish.

This manuscript is a resubmission of an earlier submission. The following is a list of the peer review reports and author responses from that submission.

Round 1

Reviewer 1 Report

The authors reported NASICON-type Na4MnCr(PO4)3 (NMCP) wrapped with reduced graphene oxide (rGO) as cathode material for sodium-ion batteries. The electrochemical performance is improved after introducing rGO and the materials are well characterized. However, before consideration for publication in journal of Polymers, the following issues should be carefully addressed.

1. English language of the manuscript needs further polishing.

2. Why the authors selected NMCP? What is the advantage compared with other NASICON-based materials? Please add related content in the Introduction part.

3. What is dosage of rGO when preparing NMCP/rGO? Was the pristine NMCP prepared under the same condition as NMCP/rGO except that without the addition of rGO? It should be stated in the Experimental part.

4. Supporting Information including Fig. S2, Table S1, Table S2, Fig. S3, Fig. S4 and Table S3 are not mentioned in the main text.

5. What is the carbon content in the pristine NMCP and NMCP/rGO? TG analysis and Raman test should be added.

6. The authors calculated the Na+ diffusion coefficient (D) based on CV and EIS test, but there is a big difference in the D value obtained from these two methods. Why? And which one is more accurate?

7. What is the performance of NMCP/rGO with different rGO content? Did the author study or optimize it?

8. Electrochemical performance comparison with previously reported NMCP based materials is necessary.

Author Response

Reviewer 1

The authors reported NASICON-type Na4MnCr(PO4)3 (NMCP) wrapped with reduced graphene oxide (rGO) as cathode material for sodium-ion batteries. The electrochemical performance is improved after introducing rGO and the materials are well characterized. However, before consideration for publication in journal of Polymers, the following issues should be carefully addressed.

  1. English language of the manuscript needs further polishing.

Response:

Thank you for reviewer’s suggestion. The English of the manuscript have been polished before resubmission.

  1. Why the authors selected NMCP? What is the advantage compared with other NASICON-based materials? Please add related content in the Introduction part.

Response:

 The electrochemical properties of NMCP had good thermal stability and high charge-discharge platform, and they were benefit to the insertion and de-insertion of sodium ions, which might be a potential cathode material for Na-ion batteries. The corresponding description have been added in the introduction part.

  1. What is dosage of rGO when preparing NMCP/rGO? Was the pristine NMCP prepared under the same condition as NMCP/rGO except that without the addition of rGO? It should be stated in the Experimental part.

Response:

The optimal rGO content in this study was 5 wt%. Yes. The pristine NMCP prepared was under the same condition as NMCP/rGO. We have mention it in the revised experimental part.

  1. Supporting Information including Fig. S2, Table S1, Table S2, Fig. S3, Fig. S4 and Table S3 are not mentioned in the main text.

Response:

Thank you for reviewer’s comments. We have mentioned Fig. S2, Table S1, Table S2, Fig. S3, Fig. S4 and Table S3 in the revised manuscript.

  1. What is the carbon content in the pristine NMCP and NMCP/rGO? TG analysis and Raman test should be added.

Response:

Thank you for reviewer’s suggestions. Indeed, we need to use TG analyses to determine the carbon content of pristine NMCP and NMCP/rGO. As shown in the TG analyses, the carbon content (rGO) was about 5 wt.%.

Fig. S7 TG analyses of pristine NMCP and NMCP/rGO composite under a heat rate of 5 oC/min in air.

  1. The authors calculated the Na+diffusion coefficient (D) based on CV and EIS test, but there is a big difference in the D value obtained from these two methods. Why? And which one is more accurate?

Response:

Thank you for reviewer’s comments. The big differences of Na+ diffusion coefficient between CV and EIS tests are due to the different principles. The apparent diffusion coefficient values calculated from CV is by change the scanning rate from 0.1 mV/s to 0.5 mV/s from 1.4V to 4.3V. The diffusion coefficient values EIS measurement was conducted in the fix voltage by changing the frequency. Sometimes, diffusion coefficient values determined by EIS measurements was more accurate than that calculated by CV because the voltage window issue. 

  1. What is the performance of NMCP/rGO with different rGO content? Did the author study or optimize it?

Response:

We tested 5, 10 and 20 wt% rGO wrapped NMCP. The following figures are the corresponding electrochemical performances. Based on these results, 5 wt% rGO was the best content. 

Fig. Pristine NMCP and NMCP/rGO with different rGO content (a) C-rate (b) Cycle life.

  1. Electrochemical performance comparison with previously reported NMCP based materials is necessary.

Response:

Thank you for reviewer’s good suggestion. We have compared the previously reported NMCP based materials in the revised manuscript.

Reviewer 2 Report

Journal: Polymers (ISSN 2073-4360)

Manuscript ID: polymers-1923209

Type: Article

Title: Synthesis and Characterizations of Na4MnCr(PO4)3/rGO as NASICON-type Cathode Materials for Sodium-ion Batteries.

Authors: Bing-Hsuan Hsu, Wei-Ren Liu*.

a)           Abstract: Expand the abstract to include all the acquired findings.

b)          Introduction: In the literature, add more than three to the number of authors who have worked on their subject and achieved results.

c)           Write the objective of the present work clearly.

d)          Why the authors didn’t measure the grain size using XRD for the samples?

e)           Why the author didn’t measure the EDX for the sampls?

f)            For mechanical properties, why the author didn’t measure the absorption of the water for the samples with time, impact and hardness, flexural and Hardness for the samples, Also the thermal conductivity of them?

g)          With the frequency, the author can measure the dielectric constants for the samples.

h)          For references, choose recent refs. Please, refer to these refs.

DOI: https://doi.org/10.1088/1742-6596/1795/1/012052

DOI: https://doi.org/10.1088/1742-6596/1795/1/012059

Best Regards

Author Response

Reviewer 2

  1. a)Abstract: Expand the abstract to include all the acquired findings.

Response:

Thank you for reviewer’s suggestions. We have expanded the abstract to include all the acquired findings in the revised manuscript.

  1. b)Introduction: In the literature, add more than three to the number of authors who have worked on their subject and achieved results.

Response:

Thank you for reviewer’s comments. We have added more than three literatures about the similar subjects in the revised manuscript.

  1. c)Write the objective of the present work clearly.

Response:

Thank you for reviewer’s comment. We will write the objective of the present work clearly in the revised manuscript.

  1. d)Why the authors didn’t measure the grain size using XRD for the samples?

Response:

Yes. The grain sizes of pristine NMCP and NMCP/rGO were 43 nm and 33 nm, respectively, by using Scherrer equation. The corresponding data and discussion have been added in the revised manuscript.

  1. e)Why the author didn’t measure the EDX for the sampls?

Response:

Reviewer’s might make a mistake. EDX mapping is shown in Fig. 4(c).

Figure 4. (a-b) TEM images of NMCP/rGO and SAED (c) NMCP/rGO elemental mappings of oxygen (yellow), phosphorus (purple), sodium (red), manganese (green), chromium (blue), carbon (orange).

  1. f)For mechanical properties, why the author didn’t measure the absorption of the water for the samples with time, impact and hardness, flexural and Hardness for the samples, Also the thermal conductivity of them?

Response:

Reviewer’s might make a mistake. We don’t need to measure the absorption of the water for the samples with time, impact and hardness, flexural and Hardness for the samples. Also, we don’t need to measure the thermal conductivity. The topic of our manuscript is about Na ion battery.

  1. g)With the frequency, the author can measure the dielectric constants for the samples.

Response:

Reviewer’s might make a mistake. We don’t need to measure the dielectric constants for the samples. The topic of our manuscript is about Na ion battery.

  1. h)For references, choose recent refs. Please, refer to these refs.

DOI: https://doi.org/10.1088/1742-6596/1795/1/012052

DOI: https://doi.org/10.1088/1742-6596/1795/1/012059

Response:

Thank you for reviewer’s suggestions. We had added these two important references in the revised manuscript.

[A1] Rasheed M.; Shihab S; Sabah O. W. An investigation of the Structural, Electrical and Optical Properties of Graphene-Oxide Thin Films Using Different Solvents. J. Phys.: Conf. Ser. 2021, 1795, 012052.

[A2] Abbas M. M.; Rasheed M. Solid state reaction synthesis and characterization of Cu doped TiO2 nanomaterials. J. Phys.: Conf. Ser. 2021, 1795, 012059.
